# Seasonality in Synergism with Multi-Pathogen Presence Leads to Mass Mortalities of the Highly Endangered *Pinna nobilis* in Greek Coastlines: A Pathophysiological Approach

**DOI:** 10.3390/microorganisms11051117

**Published:** 2023-04-25

**Authors:** Athanasios Lattos, Konstantinos Feidantsis, Ioannis A. Giantsis, John A. Theodorou, Basile Michaelidis

**Affiliations:** 1Laboratory of Animal Physiology, Department of Zoology, Faculty of Science, School of Biology, Aristotle University of Thessaloniki, GR-54124 Thessaloniki, Greece; lattosad@bio.auth.gr; 2Department of Animal Science, Faculty of Agricultural Sciences, University of Western Macedonia, GR-53100 Florina, Greece; igiantsis@uowm.gr; 3Department of Fisheries & Aquaculture, University of Patras, GR-23200 Mesolonghi, Greece; jtheo@upatras.gr

**Keywords:** *Pinna nobilis*, apoptosis, autophagy, inflammation, cell protection, mass mortalities

## Abstract

Mortalities of *Pinna nobilis* populations set at risk the survival of the species from many Mediterranean coastline habitats. In many cases, both *Haplosporidium pinnae* and *Mycobacterium* spp. are implicated in mass mortalities of *P. nobilis* populations, leading the species into extinction. In the context of the importance of these pathogens’ role in *P. nobilis* mortalities, the present study investigated two Greek populations of the species hosting different microbial loads (one only *H. pinnae* and the second both pathogens) by the means of pathophysiological markers. More specifically, the populations from Kalloni Gulf (Lesvos Island) and from Maliakos Gulf (Fthiotis), seasonally sampled, were chosen based on the host pathogens in order to investigate physiological and immunological biomarkers to assess those pathogens’ roles. In order to determine if the haplosporidian parasite possesses a major role in the mortalities or if both pathogens are involved in these phenomena, a variety of biomarkers, including apoptosis, autophagy, inflammation and heat shock response were applied. The results indicated a decreased physiological performance of individuals hosting both pathogens in comparison with those hosting only *H. pinnae*. Our findings provide evidence for the synergistic role of those pathogens in the mortality events, which is also enhanced by the influence of seasonality.

## 1. Introduction

The endemic Mediterranean Sea bivalve *Pinna nobilis* (Linnaeus, 1758) represents the largest bivalve in the Mediterranean Sea reaching up to 120 cm and inhabiting mostly soft bottom areas with vegetation (*Posidonia oceanica* and *Cymodocea nodosa*) at depths between 0.5 m and 60 m [1,2]. Anthropogenic pressures such as illegal fishing, habitat destruction and commercialization of the species decreased the populations in the Mediterranean Sea and established it as a critically endangered species on the IUCN red list [3,4,5]. On the other hand, decreases in *P. nobilis* populations continue to spread and devastate populations around the Mediterranean coastlines. The latter confirmed the presence of a haplosporidian parasite as the etiological agent of the mortalities in 2016 [6,7,8]. Thereafter, Carella et al. [9] demonstrated the presence of Mycobacteria as an extra etiological agent in the collapse of the populations in French and Italian habitats of the species at moribund populations. Moreover, *Vibrio* spp. were implicated in *P. nobilis* mortalities in lower temperatures compared to the temperatures that *Haplosporidium pinnae* infests individuals [10,11,12,13,14]. Finally, *Vibrio mediterranei* presence was implicated in the mortalities of individuals originating from Maliakos Gulf, Greece, parallel to increasing seawater temperature [12]. The latter observation was also confirmed by the presence of *V. mediterranei,* being the causative agent of mortality in stabled individuals in Italy [15]. From individuals of *P. nobilis* hosting several etiological agents emerged a discussion about the severity of each disease agent. The severity and degree of implication of multiple pathogens in the species have confirmed the synergistic effects of these agents [16].

It has been shown that infections with protozoan parasites induce immune responses in marine bivalves regardless of the effects of the thermal stress [17]. Specifically, mussels infected with the protozoan parasite *Marteilia refringens* exhibited increased inflammation, as well as apoptotic, responses [17]. However, marine invertebrates lack adaptive immune response and they solely rely on innate immune responses in order to encounter invading pathogens [18]. During potential pathogens’ invasion, the bivalve innate immune system exhibits strong responses through pattern recognition receptors and triggers the downstream immune signaling pathways by the recognition of pathogen-associated molecular patterns [18,19,20]. At the time apoptosis or host cell death constitutes a fundamental immune defense process involved in cellular homeostasis for both invertebrates and vertebrates and can be also triggered by both external and internal factors [21,22]. The intrinsic pathway is activated through the signal release by mitochondria, followed by the activation of apoptotic proteins [23]. Although the main benefits of apoptosis occur in an uninfected organism, apoptosis’ detrimental effects can be triggered by parasitic infection, depending upon the specific host-parasite situation [24]. On the other hand, the extrinsic pathway is activated upon the stimulation of specific receptors for cell death and finally the activation of caspases as the last process of the stage [23]. Furthermore, ubiquitination and autophagy are fundamental mechanisms in numerous cell processes such as cell survival, cell differentiation and removal of harmful proteins through the proteasome [16,25]. It is now clear that the autophagic pathway also eliminates several pathogens, such as viruses, bacteria and parasites (a process known as xenophagy) [26,27,28,29]. Additionally, heat shock proteins (Hsps) are responsible for homeostasis processes as well as in assisting the stimulation of cytokines, such as interleukins (Ils) and tumor necrosis factor (TNF) secretion [30]. Moreover, it is well established that Hsps play fundamental roles in response to various biotic stresses (pathogens such as parasites and parasitoids) [31,32,33,34,35].

Thus, the main objective of this study was to investigate those fundamental physiological processes in two different tissues of *P. nobilis* populations during the ambient seawater temperature increase in Greek coastlines and to compare individuals hosting both major pathogens (haplosporidium parasite and *Mycobacterium* spp.) with those hosting only the haplosporidium parasite.

## 2. Materials and Methods

### 2.1. Animal Sampling

The current research is a continuation of a previously published research [12], which reported the presence of *Vibrio mediterranei* in Fthiotis (Maliakos Gulf) in an attempt to evaluate the health status in one of the remaining surviving populations of *P. nobilis* on Greek coastlines. All sampled individuals were of similar length (42–50 cm). In the aforementioned research, two efforts of sampling were carried out in Fthiotis and in Kalloni Gulf, Lesvos Island, during February, April, June and July 2020, in compliance with the terms of the license received from the Greek Ministry of Environment and Energy (code: MEE//GDDDP89926/1117). From each sampling, five (n = 5) different animals were collected. From each animal, the mantle and gills were dissected. In the sampling during June, five moribund individuals were collected from each site at a mean seawater temperature of 24 °C and 22 °C in Fthiotis and Lesvos Island, respectively (Figure 1). Three weeks later, during the next sampling in July, both populations were found to be collapsed at average seawater temperatures of 26 °C and 24.5 °C in the aforementioned sites, respectively, and no alive individuals were observed during the scuba diving efforts. Individuals from the sampling in June were aseptically dissected and each tissue was separately placed in sterile 1.5 mL tubes and stored in liquid nitrogen for further analysis. Additionally, a small part of the digestive gland was fixed and kept for histological analysis. It should be highlighted that in Lesvos Island, where only *Mycobacterium* spp. was detected, several *P. nobilis* populations suffered from mass mortality events. However, the remaining populations reproduced, and these populations survived during the following years. On the other hand, Fthiotis populations, infected by both *Mycobacterium* spp. and *H. pinnae*, suffered a 100% mortality rate, with these populations being diminished in this particular habitat–sampling area. In line with this, no surviving populations have been observed in the following efforts in Fthiotis marine area. Measurements of surface seawater temperature were performed using a Multiparameter Water Quality Meter (Model WQC-24, DKK-TOA Company, Tokyo, Japan). In addition to water temperature, salinity, dissolved oxygen concentration and pH were also recorded.

### 2.2. Pathogen’s Molecular Detection

Approximately 20 mg of the digestive gland of each collected specimen was subjected to DNA extraction. Total DNA was extracted using the DNAEasy Blood and Tissue kit (QIAGEN, Hilden, Germany) according to the manufacturer’s instructions. The quality and quantity of the isolated DNA were evaluated in a Q5000 microvolume spectrophotometer (Quawell Technology Inc., Thmorgan, Hong-Kong, China). PCR reactions for *Mycobacterium* sp. and *H. pinnae* detection in the extracted DNA were applied as described in Lattos et al. [5].

### 2.3. SDS-Page/Immonoblot

Tissue samples preparation’ for SDS-PAGE/immunoblot analysis is described in detail by Feidantsis et al. [36]. Herein, 50 μg of protein was separated with 10% acrylamide and 0.275% bisacrylamide (*w*/*v*) or 15% acrylamide and 0.33% bisacrylamide (*w*/*v*) slab gels. Thereafter, separated proteins were electrophoretically transferred onto nitrocellulose membranes (0.45 μm, Schleicher and Schuell, Keene, NH, USA), which were then incubated with the following antibodies according to manufacturer’s instructions: anti-Hsp70 (H5147, Sigma, Darmstadt, Germany), anti-Hsp90 (H1775, Sigma, Darmstadt, Germany), anti-Bcl-2 (7973, Abcam, Cambridge, MA, USA), anti-Bax (2772, Cell Signaling, Beverly, MA, USA), anti-p62/SQSTM1 (5114, Cell Signaling, Beverly, MA, USA), anti-IL-6 (CSB-PA06757A0Rb, Cusabio, Houston, TX, USA), anti-TNF-α (CSB-PA07427A0Rb, Cusabio, Houston, TX, USA) and anti-β-actin (3700, Cell Signaling, Beverly, MA, USA). The latter was used for normalization (standard protein). The blots were washed with TBST, incubated with horseradish peroxidase-linked secondary antibodies and washed again with TBST. Thereafter, enhanced chemiluminescence (Chemicon, Illinois, USA) with exposure to Fuji Medical X-ray films was employed in order for the blots to be detected. Laser-scanning densitometry (GelPro Analyzer Software 32, GraphPad, San Diego, USA) was employed for films’ quantification.

### 2.4. Dot Blot Analysis

In mantle and PAM samples, a dot blot apparatus was employed for the determination of cleaved caspases and ubiquitin conjugate levels. Samples were diluted in a saline solution in a concentration of 5 μg mL^−1^ (150 mM NaCl). Subsequently, equal sample volumes (100 μL) were loaded in a dot blot vacuum apparatus (BioRad) and gravity-fed through a pre-soaked nitrocellulose membrane (0.45 μm), which was thereafter blocked at room temperature for 30 min with 5% (*w*/*v*) non-fat milk in TBST (20 mM Tris-HCl, pH 7.5, 137 mM NaCl, 0.1% (*v*/*v*) Tween 20). The derived nitrocellulose membrane was then incubated with the following antibodies according to the manufacturer’s instructions: anti-cleaved caspase antibody (8698, Cell Signaling, Beverly, MA, USA) and anti-ubiquitin antibody (3936, Cell Signaling, Beverly, MA, USA). The dots were washed with TBST, incubated with horseradish peroxidase-linked secondary antibodies and washed again with TBST. Thereafter, enhanced chemiluminescence (Chemicon) with exposure to Fuji Medical X-ray films was employed in order for the dots to be detected. Laser-scanning densitometry (GelPro Analyzer Software, GraphPad, USA) was employed for films’ quantification.

### 2.5. Statistical Analysis

In order to test for *p* < 0.05 level significance between experimental groups, one-way analysis of variance (ANOVA) (GraphPad Instat 3.0) and Bonferroni post-hoc test were used. Since in the present study n = 5 individuals were employed from each group, Friedman’s non-parametric test and Dunn’s post-test were also employed. The latter were applied considering the fact that normality tests have little power to test data homogeneity of small sample sizes.

## 3. Results

### 3.1. Temperature

Figure 1 depicts the seasonal seawater temperature records in Lesvos Island and Fthiotis. Although temperature was fairly similar in both sampling sites during the samplings conducted in February and April, and in June and July, seawater temperature in Fthiotis had increased compared to Lesvos Island (24.5 °C and 26.5 °C in June and July, respectively, compared to 21.5 °C and 24.5 °C). Seawater values of salinity, dissolved oxygen concentration and pH did not seasonally statistically differ.

### 3.2. Pathogen’s DNA Identification

As seen in Table 1 Mycobacterium spp. was found in all seasonal samples from Lesvos Island and Fthiotis. On the other hand, *H. pinnae* was only found in specimens from Fthiotis from February to June.

### 3.3. Heat Shock Response

In general, an increasing trend of Hsp70 and Hsp90 levels was observed towards the spring (April) and summer (June) months along with the increasing seawater temperature. Regarding Hsp70 (Figure 2A), in the mantle of Lesvos individuals, these levels increased parallel to the increasing seawater temperature, exhibiting their highest levels in June. On the other hand, the same tissue of Fthiotis individuals exhibited their highest Hsp70 levels in April, and thereafter these levels dropped even below the ones observed in February (Figure 2A). Differences between the two sampling sites were observed only in February (with Fthiotis being statistically higher than Lesvos) and in June (with Lesvos being statistically higher than Fthiotis). A different pattern was observed regarding Hsp70 levels in gills. Specifically, in the gills of individuals collected from Fthiotis, Hsp70 levels remained unchanged, while Lesvos individuals exhibited statistically increased levels only in the sampling of June (Figure 2A). Differences between the two sampling sites were observed throughout the seasonal samplings, with Lesvos being statistically higher than Fthiotis.

Hsp90 levels in the mantle of both Lesvos and Fthiotis individuals exhibited their highest levels in April, and thereafter in June, they decreased to levels similar to those observed in February. Regarding statistical differences between the sampling sites, Lesvos individuals exhibited in general higher Hsp90 levels compared to Fthiotis individuals (Figure 2B). In the gills, Hsp90 levels exhibited an increasing pattern parallel to the increasing seawater temperature, with the highest levels observed in June. Similar to the mantle, Lesvos individuals exhibited statistically higher Hsp90 levels in general compared to Fthiotis individuals (Figure 2B).

### 3.4. Apoptosis

Regarding indicators of apoptosis, Bax/Bcl-2 ratio and cleaved caspases were examined herein (Figure 3).

In many systems, members of the Bcl-2 family (Bcl-2 as an anti-apoptotic and Bax as a pro-apoptotic member) modulate apoptosis, with the Bax/Bcl-2 ratio serving as a rheostat to determine cell susceptibility to apoptosis [37,38]. Therefore, increased Bax/Bcl-2 ratio together with caspases serve as potent indicators of increased apoptotic procedure [39]. Bax/Bcl-2 ratio levels in both mantle and gills of individuals originating from Lesvos and Fthiotis exhibited an increasing trend parallel to the increasing temperature, with the highest levels observed in June (Figure 3). Regarding statistical differences between the sampling sites, Lesvos individuals exhibited statistically higher Bax/Bcl-2 ratio levels compared to Fthiotis individuals (Figure 3).

The same pattern was observed regarding cleaved caspases levels in the gills of *P. nobilis* (Figure 4). However, the mantle exhibited a different pattern: while Lesvos individuals exhibited no statistically significant differences between the seasonal sampling regarding cleaved caspases levels, Fthiotis individuals exhibited their highest cleaved caspases levels in February and June, and the latter were statistically higher compared to the ones of Lesvos in the respective samplings (Figure 4).

### 3.5. Autophagy

Regarding indicators of autophagy, SQSTM1/p62 and ubiquitin conjugates levels were examined herein (Figure 5).

During autophagy, SQSTM1/p62 protein bodies, which usually occur within autophagosomes and lysosomal structures, recruit ubiquitylated cytosolic cargo material to the autophagic isolation membrane, where it is decomposed. Inhibition of autophagy leads to an increase in the size and number of SQSTM1/p62 bodies [40]. In the gills of individuals from both Lesvos and Fthiotis, ubiquitin conjugates levels increase with increasing seawater temperature and peak in the sampling of June (Figure 5A,B). A similar indication of autophagic activity in the gills was also observed in parallel to the above decreasing SQSTM1/p62 levels (Figure 5A,B). Regarding both autophagic procedures in the gills, it seems that Lesvos individuals are more potent compared to Fthiotis ones. In the mantle of Fthiotis and Lesvos individuals, a different pattern was observed compared to the gills. Specifically, ubiquitin conjugates levels in Fthiotis exhibit their highest levels in February and June, while Lesvos individuals exhibited an increasing trend parallel to the increasing seawater temperature. In the mantle, SQSTM1/p62 levels in Fthiotis individuals exhibited no seasonal changes compared to Lesvos individuals, where decreasing levels exhibited increased autophagy parallel to the increasing seawater temperature (Figure 5A,B).

### 3.6. Inflammation

Both of the inflammatory indicators examined herein exhibited an increasing pattern in their levels parallel to the increasing seawater temperature. Specifically, TNF-α levels in the gills of both Lesvos and Fthiotis individuals, as well as in the mantle of Fthiotis individuals, depicted this pattern. On the other hand, the mantle of Lesvos individuals depicted no seasonal differences regarding TNF-α levels (Figure 6A). Regarding statistical differences between the sampling sites, Lesvos individuals exhibited statistically higher TNF-α levels compared to Fthiotis individuals (Figure 6A). As far as Il-6 is concerned, its levels increased parallel to the increasing seawater temperature in both the gills and the mantle of Fthiotis and Lesvos individuals, with Fthiotis exhibiting statistically significant higher Il-6 levels in general compared to Lesvos.

## 4. Discussion

Population declines of *P. nobilis* in the Greek coastline due to disease mortality events resulted in extended modifications (critical losses and extinction of high ecological importance) of the species’ habitat abundance. Initially, *H. pinnae* has been detected in moribund specimens, and thus it constituted the first proposed etiological agent of the Mediterranean Sea mortalities [5,6,7,8]. Other highly important bivalve pathogens such as *Vibrio* spp. and *Mycobacterium* spp. during mass mortality events in the Mediterranean Sea were also detected [5,9,10,12,41]. Although several etiological agents have been extensively described by many researchers in the last years, no literature exists, to our knowledge, on the direct impact of a rapid and long-term raise of temperature as a stress factor in the species. Critical declines, even extinction, from highly important habitats such as Thermaikos Gulf (North Greece) have been monitored during the period of extensive temperature raise during the autumn and winter of 2022. Moreover, high mortality of *P. nobilis* juveniles was also detected in both Chalkididi and Vistonikos Gulf (North Aegean Sea) during extensive periods of high temperature [17].

Thermal stress constitutes a highly important stress factor for marine bivalves and has been linked several times with disease phenomena [17]. Increased temperature has been correlated with induced antioxidant enzyme activity, hyperoxic, hypoxic and ischemia-reperfusion injuries, iron overload and intoxication in marine invertebrates [26,42,43,44,45]. The above could be attributed to the fact that thermal stress enacts a key role in the suppression of many physiological and immunological processes in marine bivalves [16,17]. Except for Hsps’ pivotal role in the organisms’ endurance to thermal stress, they are also highly important mediators in the immunity of several hosts during disease phenomena [46,47]. Our results exhibited increased levels of Hsp70 and Hsp90 levels during the highest seawater temperature observed in individuals originating from Lesvos Island, while Fthiotis individuals exhibited an increasing pattern parallel to the increasing seawater temperature, and in June these levels decreased. This decrease coincided with the presence of both pathogens (*H. pinnae* and *Mycobacterium* spp.), probably affecting the total health conditions, the immunity responses and the overall physiological performance of the animals under this synergistic infection. This observation has also been reported by Lattos et al. [16]. Similar to the above, Deane et al. [48] observed decreased Hsp90 levels in seabream (*Sparus sarba*) when challenged with the presence of pathogenic microorganisms. On the other hand, marine bivalves exposed to both increased temperatures and bacterial pathogens exhibited a higher expression of Hsp90 compared to the ones exposed solely to one stress factor [49]. Regarding parasitic phenomena, Rinehart et al. [33] showed that in the pupae of the *Sarcophaga crassipalpis* fly, Hsp23 and Hsp70 levels were significantly increased after envenomation by the endoparasitoid, *Nasonia vitripennis*. Moreover, Shim et al. [34] reported that small Hsps and *Hsc70* expression levels in *Plodia interpunctella* larvae increased in response to *Habroacon hebetor* parasitism. Similarly, *Ectomyelois ceratoniae* carob moth Hsp70 transcripts increased during *Habroacon hebetor* wasp parasitism [35]. These inconsistencies could be attributed to the Hsps’ versatile roles. Specifically, it is now well established that Hsp90 participates in multiple cellular functional processes, including also its involvement in hormonal signal transduction, cell differentiation and proliferation, stress response and apoptosis [50,51].

Concerning the apoptotic indicator Bax/Bcl-2, it demonstrated a similar pattern to that demonstrated for the heat shock response (HSR). Likewise, the synergistic effect of temperature increase and the second pathogen infection (Fthiotis individuals) downregulated the apoptotic pathway response. Particularly, the results of the present study, which exhibited that individuals hosting only *Mycobacterium* spp. (Lesvos Island individuals) expressed higher Bax/Bcl-2 levels and cleaved caspases levels, are in line with the ones by Lattos et al. [16,17]. These all indicate that increased ambient temperature [52] as well as the presence of multiple pathogenic stressors may affect this stage of apoptosis. However, and contrariwise to the above, only in the mantle tissue of Fthiotis individuals hosting both pathogens, cleaved caspases levels exhibited higher levels in the lowest and maximum seawater ambient temperature. To our knowledge, no information regarding the synergistic effect of multiple pathogens on the caspase activity exists so far. Nevertheless, it has been shown that multiple infectious agents may alter vascular cell function and provide a “trigger” for apoptotic events in human carotid endarterectomy specimens [53]. Still, no quantification regarding apoptosis in response to one or several pathogens was performed. Probably, the differential and tissue specific results obtained regarding caspases in Fthiotis individuals could be attributed to caspases’ several roles. Apart from their role in immunity, caspases play a fundamental role in the coordination of the execution phase in apoptosis by cleaving a variety of degraded proteins [54]. Whereas apoptotic pathways regulated by virus- and bacteria-derived proteins are well characterised at the molecular level, signaling events induced by protozoan parasites are yet to be addressed [55,56]. It is generally accepted though that apoptosis is inhibited by protozoan parasites [57]. From the above-obtained data, we can assume that cellular processes such as apoptosis may perform better in individuals hosting only *Mycobacterium* spp. Positive functions of apoptotic processes have been well established in a large number of biological contexts, mainly aiming at the induction of neighboring surviving cells’ proliferation in order to replace dying cells (“apoptosis-induced proliferation”), thus promoting stem cell activity and tissue regeneration [24]. However, it seems that this process could be downregulated due to stress stimuli overload in tissues of aquatic organisms. These results reinforce the hypothesis of multifactorial stressors in *P. nobilis* mortality events [16].

Stress effects in the present study are not limited to apoptosis but are also observed regarding the autophagic process. Although the latter possesses an anti-apoptotic role in the prevention of apoptotic cell death [40,58,59,60,61], it seems that the synergistic effect of ambient seawater temperature and pathogen infection leads to a stress overload suppressing both these pathways. Thermal stress has been shown to affect many cellular components and procedures such as autophagy in several organisms [62] including bivalves [52]. In these organisms, autophagy holds a key role as an important pathway against pathogenic circumstances [63,64,65]. Autophagy is an important host mechanism for the removal of intracellular bacteria and protozoans, in keeping with its primary function as a cytoplasmic clean-up process [66]. However, many pathogens have in parallel evolved strategies to protect themselves against autophagy or to harness the autophagy pathway’s components for their own benefit; although, the molecular details of such strategies are not well defined [67]. However, our results regarding the autophagic process are contrary to studies both concerning pathogen presence in the host and ambient seawater temperature raise [52,65]. Particularly, although SQSTM1/p62 levels in the present study exhibited differential results regarding seasonality and the examined tissue, it seems that infection with both pathogens in Fthiotis individuals results in increased SQSTM1/p62 levels, depicting a lower autophagic process compared to the individuals of Lesvos Island infected only by one pathogen. This fact may be characteristic of the organism’s incapability to deal with the stress overload *P. nobilis* is subjected to. Degradation of SQSTM1/p62 may lead to the stimulation of the ubiquitination process [68]. The pattern exhibited herein regarding ubiquitin conjugates confirms the above results (tissue specificity and seasonality), showcasing in general also lower levels of expression in multi-infected (Fthiotis) individuals compared to the Lesvos Island ones infected only with *Mycobacterium* spp.

Autophagy is closely intertwined with inflammatory and immune responses, and cytokines possibly mediate this interaction, since autophagy has been shown to regulate, and be regulated by, a wide range of proinflammatory cytokines such as Il [69,70]. Although inflammation is substantially correlated to infection, other environmental factors can also interplay with this process [71]. It is generally accepted that acute temperature changes negatively affect the welfare of marine bivalves by decreasing their overall physiological performance and immune responses [16,17,72,73]. However, the present results exhibit an increasing pattern in both Il-6 and TNF-α levels parallel to the increasing ambient seawater temperature, showcasing an immune response exerted due to environmental conditions. Although in some cases in the results presented herein, Il-6 levels in the mantle of Fthiotis *P. nobilis* individuals are higher compared to Lesvos individuals infected only by one pathogen, an increase of the ambient seawater temperature in the summer reverses this observation. The dual pathogenic infection seems to suppress this species’ immune response in general, thus decreasing this process’ complex and orchestrated role in the restoration of homeostatic balance in organisms after damage or pathogen invasion [74].

## 5. Conclusions

Our study demonstrates the importance of the presence of *H. pinnae* on Greek coastlines as populations infected by both pathogens exhibited a suppressed response regarding the cellular and physiological status of the species examined herein. Probably, this stress overload led to an overall reduced physiological fitness. Thus, it is of no surprise that populations hosting both pathogens critically suffered heavy losses and faced mortalities up to 100%. The latter could be attributed to the fact that warmer temperatures lead to more rapid parasite multiplication and, subsequently, to a higher rate of infection in the host. Similar to our results, Grau et al. [75] and Box et al. [76] found that the onset of mass mortality events is strongly associated with the detection of *H. pinnae*, which exhibits a preeminent role with respect to the other pathological agents considered. These observations are in accordance with our previous research, in which co-infection with *H. pinnae* and *Mycobacterium* spp. increases *P. nobilis* sensitivity to water temperature compared to individuals solely infected by one pathogen, as phylogenetic analysis and antioxidant profile have shown [77]. The differences regarding immunological and pathophysiological responses in the two examined tissues herein could be attributed to their distinct role: while the mantle is mainly a connective tissue, gills are due to their epithelial nature and respiratory and feeding organs in the bivalve. Although no statistically significant differences were found in seasonal seawater temperature, the consequential difference of 1.5 °C (mean value of 26 °C in Fthiotis compared to 24.5 °C in Lesvos) could have led to these site discrepancies. On the other hand, populations hosting only *Mycobacterium* spp. continue to overcome the mortalities and regenerate each year mostly by inhabiting deeper depths, where acute temperature variations are milder, and therefore pathogen proliferation is expected to be significantly lower. It should be taken into consideration that marine invertebrates can acquire mycobacteria from the environment, which survive in their digestive tract for periods depending on bacteria, host or environment variables, affecting the infection’s evolution, thus leading to clinically silent periods followed by disease progression [75,78]. Although “cause and effect” laboratory experiments could shed light on the cross-talking of all the underlying biochemical mechanisms exhibited in nature, the critically endangered status of this species does not allow such approaches. Therefore, research in the field should be continuous for a better understanding of this species’ interaction with its environment to be obtained.

## Figures and Tables

**Figure 1 microorganisms-11-01117-f001:**
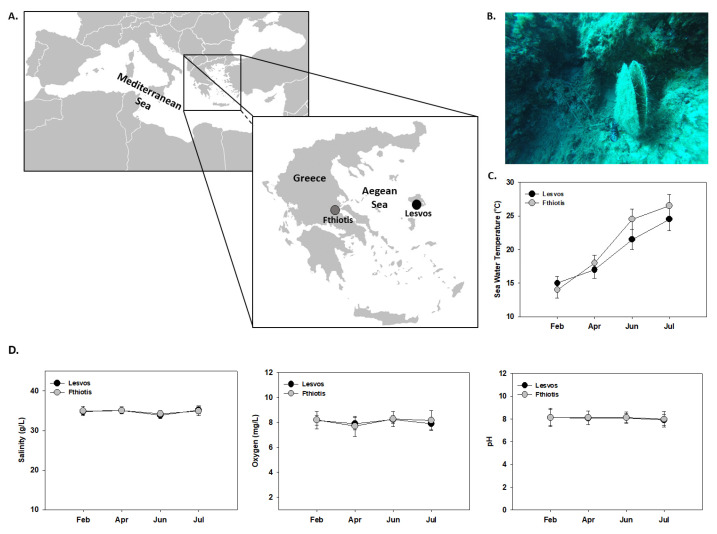
Geographic origin of specimens collected and analyzed; (**A**) natural habitat of the sampling sites (**B**), seawater temperature (**C**) and salinity (g/L), oxygen (mg/L) and pH (**D**) variations (mean ± SD) in the period from April to July 2020 during which samplings were conducted.

**Figure 2 microorganisms-11-01117-f002:**
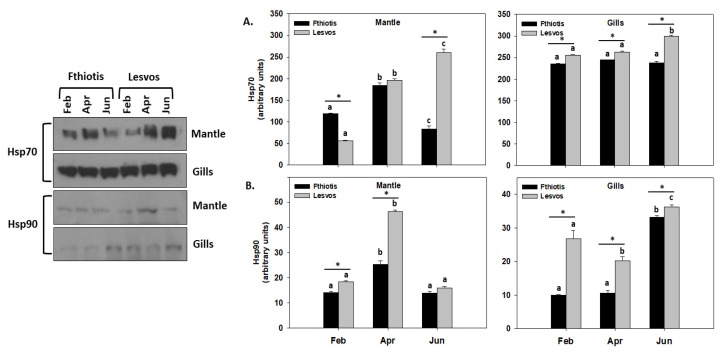
Seasonal variations of Hsp70 (**A**) and Hsp90 (**B**) levels in the mantle and gills of *Pinna nobilis* collected from Fthiotis and Lesvos Island. Values are means ± SD; n = 5 preparations from different animals. Lower-case letters depict significant differences (*p* < 0.05) between seasonal samplings, while the asterisk (*) depicts significant differences (*p* < 0.05) between sampling sites.

**Figure 3 microorganisms-11-01117-f003:**
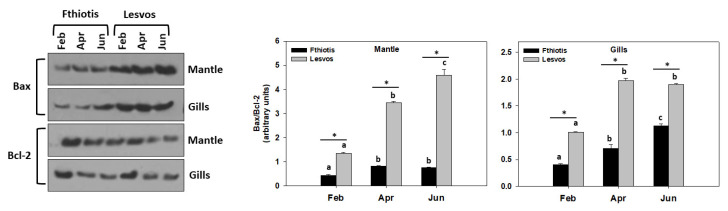
Seasonal variations of Bax/Bcl-2 ratio in the mantle and gills of *Pinna nobilis* collected from Fthiotis and Lesvos Island. Values are means ± SD; n = 5 preparations from different animals. Lower-case letters depict significant differences (*p* < 0.05) between seasonal samplings, while the asterisk (*) depicts significant differences (*p* < 0.05) between sampling sites.

**Figure 4 microorganisms-11-01117-f004:**
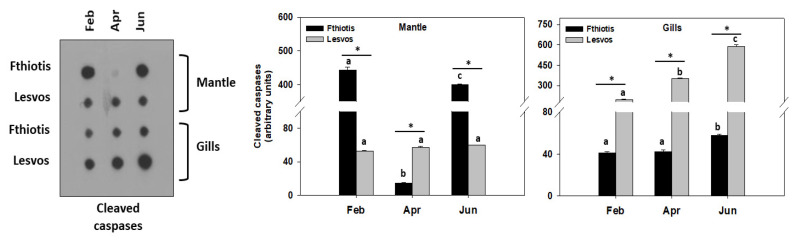
Seasonal variations of cleaved caspases levels in the mantle and gills of *Pinna nobilis* collected from Fthiotis and Lesvos Island. Values are means ± SD; n = 5 preparations from different animals. Lower-case letters depict significant differences (*p* < 0.05) between seasonal samplings, while the asterisk (*) depicts significant differences (*p* < 0.05) between sampling sites.

**Figure 5 microorganisms-11-01117-f005:**
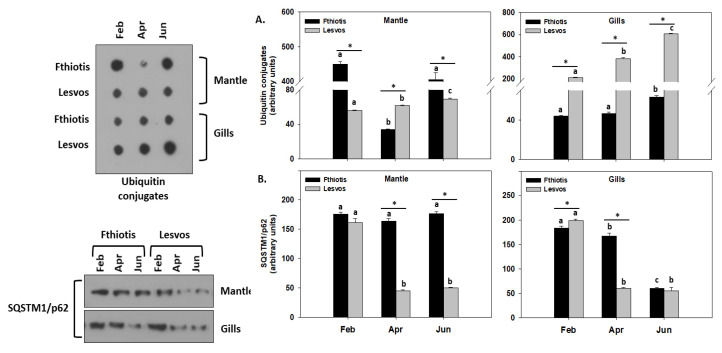
Seasonal variations of ubiquitin conjugates (**A**) and SQSTM1/p62 (**B**) levels in the mantle and gills of *Pinna nobilis* collected from Fthiotis and Lesvos Island. Values are means ± SD; n = 5 preparations from different animals. Lower-case letters depict significant differences (*p* < 0.05) between seasonal samplings, while the asterisk (*) depicts significant differences (*p* < 0.05) between sampling sites.

**Figure 6 microorganisms-11-01117-f006:**
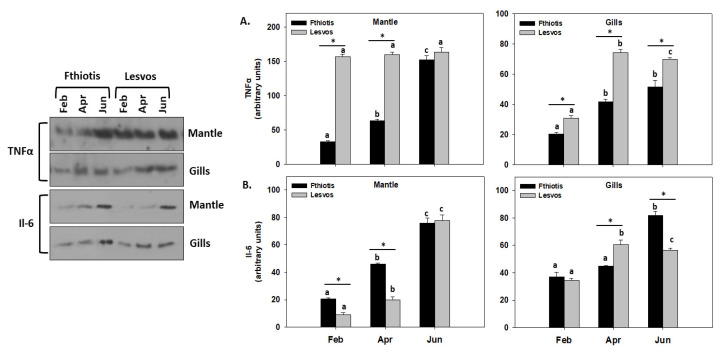
Seasonal variations of Il-6 (**A**) and TNF-α (**B**) levels in the mantle and gills of *Pinna nobilis* collected from Fthiotis and Lesvos Island. Values are means ± SD; n = 5 preparations from different animals. Lower-case letters depict significant differences (*p* < 0.05) between seasonal samplings, while the asterisk (*) depicts significant differences (*p* < 0.05) between sampling sites.

**Table 1 microorganisms-11-01117-t001:** Identification of *Mycobacterium* spp. and *H. pinnae* in the samples of *Pinna nobilis* collected from Fthiotis and Lesvos Island.

Lesvos Island Samples	*Mycobacterium* spp.	*H. pinnae*	Fthiotis Samples	*Mycobacterium* spp.	*H. pinnae*
Feb S1	+	-	Feb S1	+	+
Feb S2	+	-	Feb S2	+	+
Feb S3	+	-	Feb S3	+	+
Feb S4	+	-	Feb S4	+	+
Feb S5	+	-	Feb S5	+	+
Apr S1	+	-	Apr S1	+	+
Apr S2	+	-	Apr S2	+	+
Apr S3	+	-	Apr S3	+	+
Apr S4	+	-	Apr S4	+	+
Apr S5	+	-	Apr S5	+	+
Jun S1	+	-	Jun S1	+	+
Jun S2	+	-	Jun S2	+	+
Jun S3	+	-	Jun S3	+	+
Jun S4	+	-	Jun S4	+	+
Jun S5	+	-	Jun S5	+	+

(S represents the number of the sample, + represents presence, while - represents the absence of a pathogen).

## Data Availability

The data presented in this study are available on request from the corresponding author. The data are not publicly available due to local authorities’ privacy restrictions.

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
