# Peer review of "Seasonality in Synergism with Multi-Pathogen Presence Leads to Mass Mortalities of the Highly Endangered Pinna nobilis in Greek Coastlines: A Pathophysiological Approach"

_microorganisms, 2023, doi:10.3390/microorganisms11051117_

Round 1
Reviewer 1 Report
The article "Seasonality in synergism with multi-pathogen presence leads to mass mortalities of the highly endangered Pinna nobilis in Greek coastlines: a pathophysiological approach" brings new insights into the response to a multipathogen infection with Haplosporidium pinnae and Mycobacterium sp. The authors presented the differences between two populations of P. nobilis, one infected only with Mycoobacterium spp. in the island of Lesvos and the other infected with Mycobacterium spp. and H. pinnea in Ftihiotis. Furthermore, by connecting the same indicators with the sea temperature, new, scientifically interesting, facts for the pathophysiological response of shellfish have been established.
The article is well organized with all sections well developed, methodology clearly explained and uniform in the displayed results and therefore easy to follow. It has a slightly different approach than the previously published article with a similar topic: "Lattos A, Feidantsis K, Georgoulis I, Giantsis IA, Karagiannis D, Theodorou JA, Staikou A, Michaelidis B. Pathophysiological Responses of Pinna nobilis Individuals Enlightens the Etiology of Mass Mortality Situation in the Mediterranean Populations. Cells. 2021 Oct 22;10(11):2838. doi: 10.3390/cells10112838. PMID: 34831063; PMCID: PMC8616554."
Here are some major areas of questions to consider:
Since the article is published in the journal Microorganism, within a Parasitology section, I expected an emphasis on the influence of parasite infections (H. pinnae) on the pathophysiological response of bivalves (P. nobilis).
Although all results are logically explained and supported by references, there are alternative explanations that the authors did not consider:
1. The results do not state whether the temperatures differed significantly between the two investigated populations - (according to Figure 1. C it is evident that there are no significant differences) from which it follows that all differences between the populations can be a consequence of infection with H. pinnae. Although very clear (significant) differences between the studied populations and differences within individual populations (related to sea temperature) were shown, I expected from the Authors to connect the obtained differences with the known differences in the dynamics of infection with Mycobacterium spp. and H. pinnae (or similar parasites). Namely, it is known that warmer temperatures lead to more rapid parasite multiplication and a higher rate of infection in the host (for H. pinnea). So the results obtained in their research, especially seasonal variations in heat shock response and indicators of autophagy in mantle tissue, could be explained differently.
2. The authors compared pathophysiological parameters in two different tissues, mantle and gills. Mantle is mainly a connective tissue and gills are epithelial tissues with huge differences in immune response. Please, state the differences between tissues in the introduction and refer to them in the discussion. Furthermore, histopathological studies show that H. pinnae have tropism for the digestive gland and mantle tissue.
This study further reinforces the hypothesis of a multifactorial stressor in P. nobilis mortality events - the author's explanation for the established differences in heat shock response (L283-289) more strongly supports this hypothesis, which is only confirmed by the established downregulation effect of multiple infection on apoptosis (L294-L297), and according to results (Figure 5, A) ubiquitin conjugates.
In the Conclusions, the authors make a statement that is not visibly based on the presented data or available literature (L367-L370) which may lead the reader to wrong conclusions.
Minor issues:
Please correct the statement about study objective- it should be replaced by haplosporidium parasite with Mycobacterium spp. (L72-L75).
Overall, the article is well written and all presented data are logical and scientifically plausible but my opinion is that this article would benefit from giving emphasis on the influence on parasite infections considering alternative explanations for gathered results.
Therefore, I suggest moderate changes in discussion and conclusions section, and minor changes in the introduction section.
Reviewer 2 Report
Here you can find attacched my review.

Reviewer 3 Report
Dear Mr. Fred Gong
MDPI - Multidisciplinary Digital Publishing Institute St. Alban-Anlage 66, 4052 Basel, Switzerland
E-Mail: fred.gong@mdpi.com
Reviewer comments related to: Manuscript ID: microorganisms-2344976
Type of manuscript: Article
Title: Seasonality in synergism with multi-pathogen presence leads to mass mortalities of the highly endangered Pinna nobilis in Greek coastlines: a pathophysiological approach
Authors: Athanasios Lattos, Konstantinos Feidantsis , Ioannis . A Giantsis, John A. Theodorou, Basile Michaelidis , Submitted to section: Parasitology,
The work is an important contribution to better understand the roles of Haplosporidium pinnae and Mycobacterium spp. are in mass mortalities of P. nobilis populations in Mediterrranean sea, and in particular to verify the potential synergistic effect between them.
The manuscript looks interesting, the investigation is well set up, rigorous and complete.
The figures/tables are well made and immediate in their reading.
The bibliographic sources are numerous and pertinent.
Some considerations:
The number of individuals for each sample is low and does not allow for the application of a robust statistical calculation.
The lack of histological investigations does not allow a verification of the intensity of parasitic infection by Aplosporium as well as the possible presence of hemocytes in the foci of mycobacterial infection. Indeed it should be taken into consideration that the marine invertebrate can acquire mycobacteria from the environment, surviving in their digestive tract for long periods and the evolution of infection depends on bacteria, host or environment-dependent variables, leading to clinically silent periods followed with disease progression…please check these recent works: Tiscar et al., 2023, New insights about Haplosporidium pinnae and the pen shell Pinna nobilis mass mortality events, Journal of Invertebrate Pathology 190 (2022) 107735; Grau et al., 2022, Wide-Geographic and Long-Term Analysis of the Role of Pathogens in the Decline of Pinna nobilis to Critically Endangered Species, Frontiers, doi: 10.3389/fmars.2022.666640.
Knowing the age of the animals would have been a further correlation parameter between mycobacterial infections, generally more easily present in older animals.
Anyway congratulations, it's a good job.
I have no major revisions to propose, only the minor revisions above cited.
References:
Line 522: ….Yao, Y. ming Autophagy…. please change ming in Ming (uppercase character).
